# Nuclei segmentation of HE stained histopathological images based on feature global delivery connection network

Peng Shi[1,5]*, Jing Zhong[2], Liyan Lin[3], Lin Lin[4], Huachang Li[1,5], Chongshu Wu[1,5]

1 College of Computer and Cyber Security, Fujian Normal University, Fuzhou, Fujian, China, 2 Department of Radiology, Fujian Medical University Cancer Hospital, Fujian Cancer Hospital, Fuzhou, Fujian, China, 3 Department of Pathology, Fujian Medical University Cancer Hospital, Fujian Cancer Hospital, Fuzhou, Fujian, China, 4 Department of Radiology, Fujian Medical University Union Hospital, Fuzhou, Fujian, China, 5 Digit Fujian Internet-of-Things Laboratory of Environmental Monitoring, Fujian Normal University, Fuzhou, Fujian, China

☯ These authors contributed equally to this work.
* pshi@fjnu.edu.cn

**Data Availability Statement:** Totally three datasets are used in this research including two public open datasets and one clinical dataset. The images used in the experiments are be available in the

## Abstract

The analysis of pathological images, such as cell counting and nuclear morphological measurement, is an essential part in clinical histopathology researches. Due to the diversity of uncertain cell boundaries after staining, automated nuclei segmentation of Hematoxylin-Eosin (HE) stained pathological images remains challenging. Although better performances could be achieved than most of classic image processing methods do, manual labeling is still necessary in a majority of current machine learning based segmentation strategies, which restricts further improvements of efficiency and accuracy. Aiming at the requirements of stable and efficient high-throughput pathological image analysis, an automated Feature Global Delivery Connection Network (FGDC-net) is proposed for nuclei segmentation of HE stained images. Firstly, training sample patches and their corresponding asymmetric labels are automatically generated based on a Full Mixup strategy from RGB to HSV color space. Secondly, in order to add connections between adjacent layers and achieve the purpose of feature selection, FGDC module is designed by removing the jumping connections between codecs commonly used in UNet-based image segmentation networks, which learns the relationships between channels in each layer and pass information selectively. Finally, a dynamic training strategy based on mixed loss is used to increase the generalization capability of the model by flexible epochs. The proposed improvements were verified by the ablation experiments on multiple open databases and own clinical meningioma dataset. Experimental results on multiple datasets showed that FGDC-net could effectively improve the segmentation performances of HE stained pathological images without manual interventions, and provide valuable references for clinical pathological analysis.

Supporting information. Ineterested readers could ask the Fujian Medical University Union Hospital Ethics Committee (xhyyllwyh@163.com) for more available data.

**Funding:** This work was supported by the Fujian Science and Technology Innovation Joint Fund (2018Y9112) and the Fujian Health Science Research Talent Training Project (2019-ZQN-17). The funders had no role in study design, data collection and analysis, decision to publish, or preparation of the manuscript.

**Competing interests:** The authors have declared that no competing interests exist.

# Introduction

As one of the golden standards in clinic, analysis of stained tissue section images plays an important role in histopathological diagnosis [1]. Hematoxylin-Eosin (HE) staining is the most commonly used techniques in dealing with pathological paraffin sections, especially in analysis of tumor tissue microscopic images [1, 2], in which the nucleus is stained hyacinthine by alkaline hematoxylin, while the cytoplasm is stained red by acidic eosin. As the characteristic changes of normal cells after cancerization are mostly reflected in nuclei, the statistical results of the number, size and morphology of nuclei and other indicators can be used for cancer grading [3], which is critical for the formulation of treatment plans for patients [4].

In most of pathological image processing workflows, cell segmentation and quantification are necessary steps for precise cell structures and distributions, and eventually quantify statistical results for final identification [5]. However, HE stained images always have no obvious color differences or clear borders between different parts of numerous cells, which causes difficulties in manual or automated pathological image analysis in practice [6]. Meanwhile, colors of HE stained images may have differences between various acquisition conditions such as time, light and contrast [7], and a variety of diseases, organs, positions, and stages also have quite different cell morphologies in acquired pathological images [8]. Therefore, construction of the common and representative feature space of cell morphologies in an efficient way becomes the main concern of automated HE stained image segmentation.

As shown in Fig 1, the HE-stained pathological images vary a lot in terms of organs and imaging qualities, which need a general automatic segmentation strategy in practice. Many previous cell segmentations methods were based on traditional image processing, such as threshold determination [10], contour evolution model construction [11], and seed point marking [12], which located cell boundaries by iterative operation of pre-set features to certain terminal conditions. With the development of machine learning, classic learning-based methods such as K-means clustering [13], fuzzy C-means clustering [14] and Support Vector Machine (SVM) [15] have been applied to image segmentation, in which pixels or small patches are classified into different categories. Due to the diversity of nuclear morphology, the inhomogeneity of staining and the variability of dye quality, machine learning based methods always rely on well-designed local feature sets, and are difficult to include high representative features and lack of neighborhood receptive fields. Meanwhile, manual labeling is essential in supervised learning methods, which cannot meet efficiency requirements in clinic, and for unsupervised learning, the stability of automatic pixelwise samples selection is still a challenging step in training due to diversities discussed above.

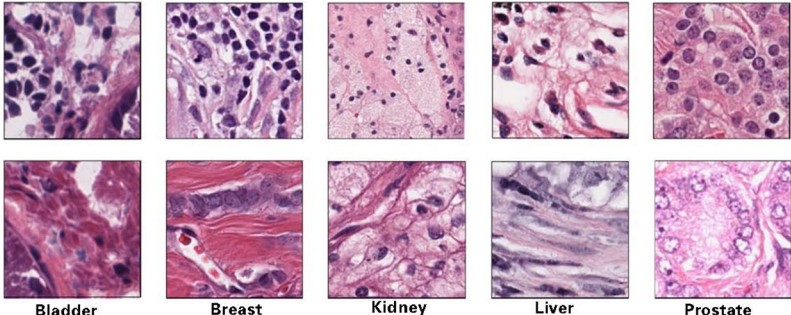

**Fig 1. Sample HE stained images from different organs in MoNuSeg [9] dataset, in which high quality stained images are shown in the first row, and the second row contains low quality samples.**

With its high precision and automatic extraction of deep features, Convolutional Neural Networks (CNNs) are widely used in current image segmentation including pathological analysis. Based on the main structure of CNN, Fully Convolutional Network (FCN) [16] uses deconvolution layers to upsample the convolution feature image, and restore it to the same size of the input image to predict the category of each pixel and complete the segmentation. As the improvement of FCN, UNet [17] includes more high resolution and classification features produced in convolutions as supplements to the upsampling directly, which highly improves the resolutions in the image restoration stage. To enhance the feature expression abilities of pathological image, researchers proposed multiple ways including the introduction of residual module, a multi-scale feature extraction module, attention mechanism, and the multi-model combination way. Li et al. [18] added a cascade residual fusion module in the decoder stage of UNet to improve detection performance in the decoding process. Zeng et al. [19] proposed RIC-UNET by adding optimization methods such as residual module, multi-scale perception and attention mechanism to segment the nucleus more accurately. Pan et al. [20] proposed a cavity depth separable convolution AS-UNet, in which the cavity convolution module was combined by cascade and parallelism, which could extract and combine multi-scale features and had better perception ability for larger or smaller nuclei. Wan et al. [21] used an improved vacuum-pyramid pooling UNet (ASPP-UNet) to capture multi-scale nuclear features and obtain their context information without reducing the spatial resolution of feature maps. Saha et al. [22] also added spatial pyramid pooling and trapezoidal long and short-term memory modules into UNet network to obtain Her2Net to retain more encoder information in decoder.

Some researchers used multi-branching and multi-model stacking methods to improve nuclear segmentation performance based on UNet structure. Navid et al. [23] proposed a spatial awareness network (SpaNet) to capture spatial information in a multi-scale manner. Double-headed and single-headed structures were designed to predict the nucleus pixel and its centroid. Zhao et al. [24] decomposed HE stained images and constructed a triple UNet network with RGB branches, HE branches and segmentation branches. The features extracted from RGB and HE branches were then fused to the segmentation branch to learn better representations. Kang et al. [25] designed a two-stage learning framework by stacking two UNets, and added nuclear boundary prediction to transform the original binary segmentation task into a two-step task, in which the first step was to estimate the kernel and its rough boundary, and the second step was to output the final fine segmentation result. In addition, Pan et al. [26], in the stage of training set making, adopted the sparse reconstruction method to initially remove the background and highlight the nuclear region.

Considering that the nucleus is stained blue and purple, the cytoplasm is stained red, and the unstained Extra Cellular Space (ECS) appears white, the difficulty of pathological image segmentation lies not in the recognition of the boundary between the cell body and the ECS, but in the recognition of the boundary between the nucleus and the cytoplasm. Therefore, the segmentation model design should be optimized to fully consider the recognition of the nucleus boundaries. In addition, the convolution is difficult to obtain global features of the input image due to its limited receptive field, and the relationship between feature channels of the same layer cannot be obtained due to its singleness and locality. Therefore, proper extension of receptive fields and information transfer between adjacent layers needs to be considered in designing network architectures. In this paper, we propose a fully automated processing pipeline to analyze HE stained images without human intervention. The main contribution of our unsupervised approach includes a Full Mixup training sample generation strategy based on asymmetric labels and HSV color transform, a dynamic training workflow and a newly designed FGDC network architecture with proper extension of receptive fields

and information transfer between adjacent layers, which identifies the boundaries between the nucleus and cytoplasm with high accuracy and effectiveness. The rest of the paper is organized as follows. In Section 2, we firstly describe the HE stained images from separate sources, and then the training sample and label generation, modules design, and dynamic training of FGDC-net. In Section 3, comparative experimental results are presented to show the improvements of the proposed methods which covers both quantitative and qualitative evaluations on multiple datasets. Finally, conclusions and further improvements are discussed in the last Section.

## Materials and methods

### Public and own clinical datasets

To deal with the diversities of HE stained images more efficiently and robustly, three datasets were used in the training and testing of the proposed framework, which include open two multi-organ datasets of Kumar [8] and MoNuSeg [9], and HE stained images of meningioma from our clinical research [27].

**Kumar and MoNuSeg datasets.** Two datasets were publicly released for testing algorithms that accurately segment nuclei, in which the Kumar dataset consisted 30 HE stained pathological images acquired from 18 different hospitals, and the MoNuSeg dataset was downloaded from Medical Image Computing and Computer Assisted Intervention (MICCAI) 2018 Multi-Organ Pathological Image Nuclear Segmentation Challenge. Distribution of each organ and division in training of the two datasets are shown in Table 1.

**Own clinical meningiomas dataset.** As the second highest brain cancer risk, the overall incidence of meningioma increased by 4.6% annually in 2004–2009, and remained stable from then [28]. World Health Organization (WHO) provides a three-point overall cancer risk scale including (I) benign, (II) atypic, and (III) anaplastic or malignant meningioma, in which both the level II and III are collectively known as high-grade meningioma [29]. The tissue samples collected in the experiments were histological sections of high grade and low grade meningiomas, were all from clinical cases of Fujian Medical University Union Hospital (Ethical approval No. 2019KJTYL024). In the dyeing image acquisition stage, the macroscopic tissue sections were segmented to obtain microscopic tissue sections, and HE staining was performed using standard histological methods.

In order to ensure the comprehensiveness and diversity of the original images, images of multiple cases from clinic were included for the training process. First, there were totally 60 cases of meningiomas blindly chosen on a patient level, in which 30 cases are from high-grade and other 30 cases are low-grade. Second, one original image was randomly selected from each case, so we had 60 original images in our own dataset. Then, 10 images from each grade of cases were randomly selected as the test set, and there were totally 20 images for test. Finally,

**Table 1. Sample images from multiple datasets for image segmentation.**

|  |  | Breast | Liver | Kidney | Prostate | Bladder | Colon | Stomach | Brain | Lung | Meningioma | |
|--|--|--------|-------|--------|----------|---------|-------|---------|-------|------|------|------|
|  |  |  |  |  |  |  |  |  |  |  | High | Low |
| Kumar | Train | 4 | 4 | 4 | 4 | 0 | 0 | 0 | - | - | - | - |
|  | Test | 2 | 2 | 2 | 2 | 2 | 2 | 2 | - | - | - | - |
| MoNuSeg | Train | 6 | 6 | 6 | 6 | 2 | 2 | 2 | 0 | 0 | - | - |
|  | Test | 2 | 0 | 3 | 2 | 2 | 1 | 0 | 2 | 2 | - | - |
| Ours | Train | - | - | - | - | - | - | - | - | - | 20 | 20 |
|  | Test | - | - | - | - | - | - | - | - | - | 10 | 10 |

the remaining 40 images including half high-grade and half low-grade images formed the training set. The ratio between the training and test datasets is 2:1, and the image quantity distribution of the final dataset is also shown in Table 1.

**Golden standards generation of labeling.** In practice, one 1536×2048 meningioma pathological image averagely contains about 800 nuclei that need to be labeled, so a total of about 16,000 nuclei needs to be labeled in 20 training images. In clinical medical analysis, MaZda [30] software (V4.6) is often used to label nuclei inside the Regions of Interest (ROI) in pathological images. After labeling, the labeling results were sent to pathologist to evaluate the quality of labeling and feedback those unreasonable labeling for correction. The correction-feedback process continued until the error rate of mistaken labeled nuclear pixels in each image was less than 5%, which was an empirical value suggested by our pathologists and made the labeling results dependable as the golden standards for the following training and validations.

## Automatic generation of training sample patches and labels

Binary image label maps are essential for the training of image segmentation networks, in which small patches containing nuclei are randomly selected from original images to generate their corresponding pseudo labels. In our previous research, we have proposed a reliable training sample generation method in an unsupervised manner based on K-means clustering results, and a plain Full Mixup strategy to enhance the training sets by adding up two patches and their label maps [27].

**Construction of patches and corresponding pseudo labels.** Nevertheless, considering that the image patches of the training sample is too small to have enough local receptive fields, it is difficult to determine the category of pixels when a selected image patch belongs to the same type of cell structure. Simply increasing the size of the patches will also greatly reduce the number of samples that can be selected. To solve this, the receptive field of training sample patches are enlarged by using the padding operation based on the original size, and the size of label map remains unchanged.

As shown in Fig 2, eligible label patches are firstly selected, which are marked as green boxes in the binary pseudo label map generated by clustering of original image pixels, where

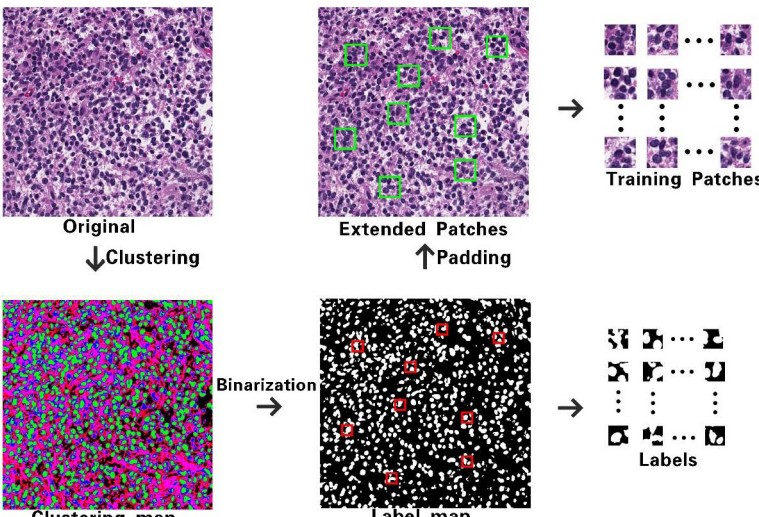

**Fig 2. Construction of sample patches and corresponding pseudo labels for training.**

three colors in the clustering map represent different tissue structures including nuclei, cytoplasm and ECS. Then the expanded image blocks after padding are captured as training samples, which are marked as red boxes in the original image. The padding factor is set as $\frac{N}{2}$ based on the label patch size of N×N, and there is a certain size difference between the training sample and its label patch. Since the training process is guided by the labels, the asymmetric model of FGDC-net focuses its attention on the central regions of the sample corresponding to its label, which plays a certain effect similar to the attention mechanism in effect.

**Mixup of image patches in HSV color spaces.** Since the training sets we constructed are composed of the most well-stained nuclei from different images, the lack of lightly stained nuclei results in weak generalization ability in dealing with different source datasets. In our previous work, a plain Full Mixup strategy was used to solve this by merging images with different weights and directly superimposes the labels of the images.

However, when the images come from different hospitals with different dyeing conditions, a large variation of images usually cause big differences between the mixed images and normal pathological images. For example, images of column one and two are from different sources and the mixed image has very light blue nuclei in the zoomed ROI of mixed image patch in column three, which may cause missing detection in the following segmentation. Common normalization cannot solve this problem as shown in the right-most histograms of blue pixels of nuclei, which have quite similar mixing result as the originals.

HSV color space [31] is composed according to the intuitive features of colors, Hue, Saturation and Value. Unlike three channels in RGB color space, HSV only uses Hue to control the variety of colors. Therefore, the range of color distribution changes in HSV space is much smaller than that in RGB space after performing the mixing operation. After normalization, the RGB images are firstly converted into HSV space as illustrated in the third row and then the blended HSV image is transformed back to RGB as the zoomed image in the same row shows. As illustrated in the distribution histograms of blue color of nuclei, more concentrated distribution of blue nuclei pixels can be found in mixed HSV-based image, suggesting the HSV-based Full Mixup generates high quality image patches for training the network. Comparisons between different mixing strategies are shown in Fig 3.

## Structure of feature global Delivery Connection Network

After the generation of training samples and labels, the segmentation network structure is also improved with the idea of attention mechanism. As shown in Fig 4, an asymmetric segmentation network named FGDC-net is proposed based on the size of automatically captured image

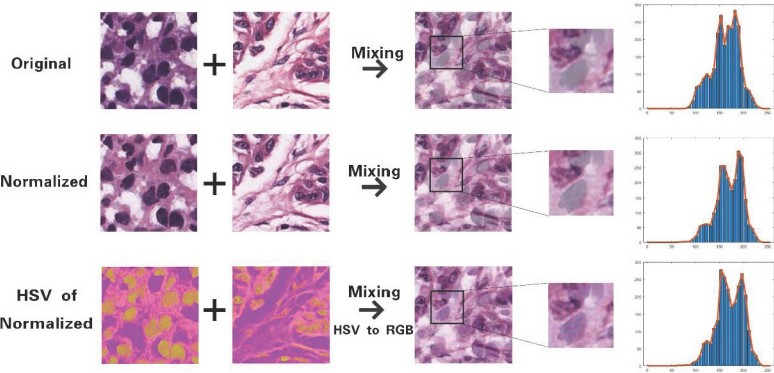

**Fig 3. Comparison between mixing based on RGB and HSV space.**

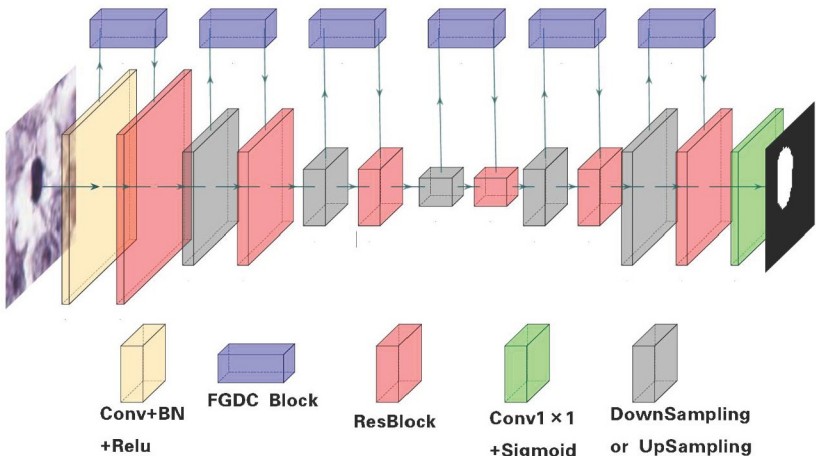

**Fig 4. Workflow of image segmentation through FGDC-net.**

sample patches of training set. As discussed above, a training patch is four times larger than its corresponding label map. Therefore, by getting through FGDC-net, the output of segmentation result only contains the central region of the input image patch, which has the same size as its corresponding pseudo label. The sizes of input (amber) and output (green) layers of FGDC-net are not equal, so as to increase the local receptive field of the central region to be segmented. In addition, FGDC-net abandons the jumping connections between codecs commonly used in UNet structures and uses FGDC modules instead to learn the relationships between feature channels at each layer and pass information selectively, which are illustrated as arrows of information flows in Fig 4.

**FGDC module.** The convolution operation is difficult to obtain the global features of the image because of its limited receptive field, and the relationship between feature channels of the same layer cannot be obtained due to its singleness and locality. In order to compensate for the disadvantages of convolution operation and increase the connections between adjacent layers, FGDC module is designed to use sigmoid-like gating to assign weights to intra-layer feature channels of each layer to achieve feature screening, and Fig 5 illustrate the implementation details.

In Fig 5, three continuous FGDC modules in a fragment of the network are marked as dark and light blue blocks. Assuming Fl is the input feature map with size of, and $S_l$ is the output of

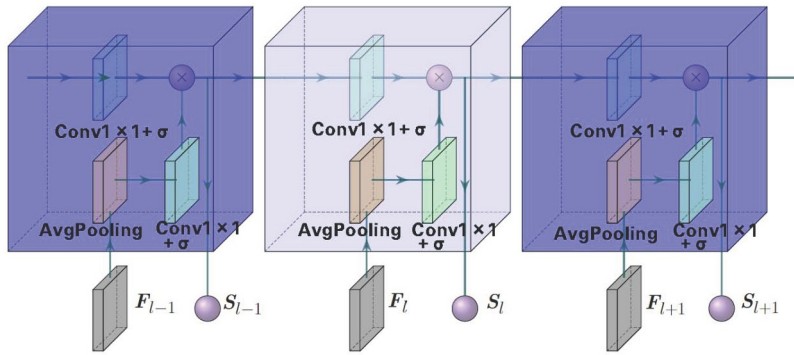

**Fig 5. Information transmission flows in FGDC modules.**

FGDC module in the $l$-th layer, the average pooling (AP) and input information of the $l$-th layer $i_l$ are calculated as below.

$$\text{AP}_1^c = \frac{1}{H \cdot W} \sum (F_l^c(:)), c \in C_l \tag{1}$$

$$i_1 = \sigma(AP_l * K_d + b_d), d = C_l \tag{2}$$

where $F_l^c$ and $AP_l^c$ are feature image and the average pooling of c-th channel in the $l$-th layer respectively, so is the total average pooling of the whole $l$-th layer. $*$ means convolution, $K_d$ is a $1 \times 1$ convolution kennel in which d is the number of kennels, b is the bias and $\sigma$ is a sigmoid activation function. The gating calculation based on $S_{l-1}$ transmitted from the $l-1$th to $l$-th layer is as follows.

$$g_1 = \sigma(S_{l-1} * K_d + b_d), d = 2C_{l-1} \tag{3}$$

It needs to be informed that $d = 2C_{l-1}$ in encoding, and $d = \frac{1}{2}C_{l-1}$ in decoding, and the output of $l$-th layer $S_l$ is $1 \times 1 \times C_l$ in size, which is the product of $g_l$ and $i_l$ after gating.

$$S_1 = g_l \cdot i_l \tag{4}$$

As the information transmission flows show, FGDC module firstly conducts gated mapping on the output of the upper layer $S_{l-1}$ to obtain the weight of the dimension matching with current layer while screening the features. Secondly, the feature map $F_l$ of the local layer is integrated and mapped to obtain the information between neighborhood layers $i_l$. Then, the upper layer information and the local layer information are integrated as the input into ResBlock module, which contains feature weights of the $l$-th layer $S_l$, and is used as the control information of the feature results obtained by convolution of this layer.

**ResBlock module.** As shown in Fig 6, the ResBlock module [32] integrates output information $S_l$ of FGDC module with feature map $F_l$ extracted from the $l$-th layer. Calculations of residuals are conducted, where represents the residual result and is the output of the ResBlock module.

$$\hat{F}_1 = K_d'' * ReLU(F_l * K_d' + b_d') + b_d', d = C_l \tag{5}$$

$$\tilde{F}_1 = ReLU(\hat{F}_l \cdot S_l + F_l) \tag{6}$$

It should be noted that $K_d'$ and $K_d''$ are different convolution kernels with size of 3×3, which are used in two different convolution processes in Eq (5).

## Optimization of training strategy

Except for above improvements in build the training set and the network structure, the training strategy of the proposed network is also optimized in two main aspects, mixed loss functions and dynamic training with flexible epochs.

**Combined loss functions.** In order to be suitable for those smaller intact or partial nuclei in the training sets of image patches, the Binary Cross Entropy (BCE) Loss and Dice loss functions are combined as the optimization objective, which are defined as follows.

$$\text{Loss} = mean(l_1, l_2, \ldots, l_N) \tag{7}$$

where

$$l_n = \frac{BCEloss_n + Diceloss_n}{2} \tag{8}$$

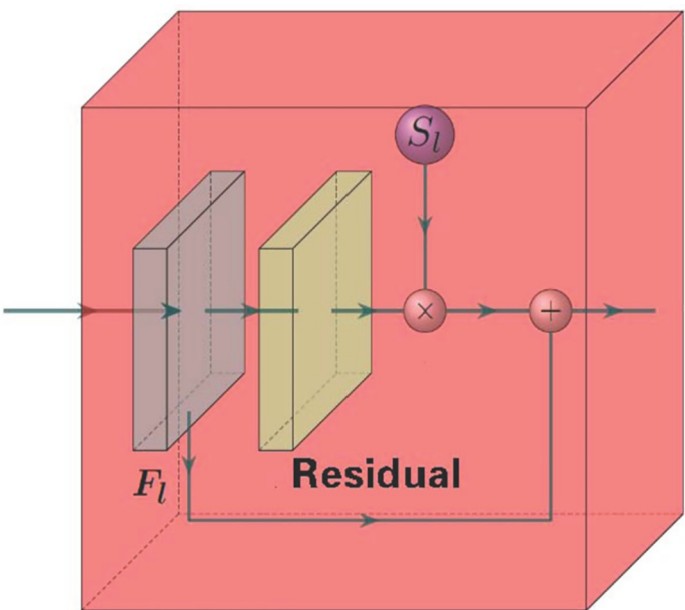

**Fig 6. Details of ResBlock module.**

in which N is the total batch size and n is a n-th batch in the training phase, $l_n$ is one training sample in each batch, $y_n$ is its corresponding label, and $x_n$ is the predicted result of category belonging to, which is within the range between 0 and 1 as non-nuclei or nuclei areas.

**Dynamic training.** To further enhance the generalization ability of the model, a dynamic training strategy with flexible epoch was proposed previously in our research, in which the algorithm dynamically modified the probability of Full Mixup by using the feedback of both Jaccard Similarity (JS) and the Dice Coefficient (DC) indexes of the validation set, and the number of epochs was determined by the increase of the probability of Full Mixup.

In the dynamic training, if the prediction ability of the model for unmixed images is higher than that of mixed images, the proportion of mixed images in the training set will be enhanced by increasing the mixing probability, and vice versa. Then, due to the randomness of the mixed image, the probability of Full Mixup gradually increases to a certain threshold and the pre-defined epoch of training will be interrupted, which makes the model adaptively allocate the number of mixed samples according to the validation set index. The probability-based adjustment learns the features of both mixed and unmixed image patches, and makes the model more flexible to the diversity of inputs pathological images acquired from various conditions.

## Results and discussions

### Building of the training datasets

To enhance the generalization ability of the proposed network, diversified training and testing datasets from multiple resources were built and divided. For Kumar dataset, training set and test set were separated according to literature [8]. For MoNuSeg dataset, the original dataset division of the segmentation challenge was followed [9]. We also splitted the training set and testing set in a 2:1 fashion for own clinical Meningioma dataset, and the details are shown in Table 2.

**Table 2. Division of original images for training and testing datasets.**

|  | Size | Training | Testing |
|---|---|---|---|
| Kumar | 1000×1000 | 16 | 14 |
| MoNuSeg | 1000×1000 | 30 | 14 |
| Meningioma | 1536×2048 | 40 | 20 |

**Table 3. Division of original images for training and testing datasets.**

|  | Total | Training | Validation | Testing |
|---|---|---|---|---|
| Kumar | 12,000+ | 8,000+ | 4,000+ | 23534 |
| MoNuSeg | 14,000+ | 10,000+ | 4,000+ | 23534 |
| Meningioma | 17,000+ | 11,000+ | 5,000+ | 110500 |

In order to minimize the influence of uneven dyeing and light sources, and obtain more available training samples automatically, we randomly captured 32 sub-graphs with the size of 500×500 from each original image for the training sample selection, and allocated training sets and verification sets with the ratio of 2:1. The numbers of final selected samples from each dataset are shown in Table 3.

## Ablation experiments

To verify the effectiveness of model improvements and optimizations in the network training, the ablation experiments mainly included two aspects, comparisons between a batch of common segmentation methods and different levels of training optimizations. In the aspect of model structure, two main improvements need to be verified. First, the feasibility of feature global transitive connection of FGDC module was compared with jump connection of Res-Block module. Second, the effectiveness of training sample padding was verified. For the improvements in the training, there are three main optimization designs, Full Mixup in HSV space, mixed loss, and dynamic training strategy. Ablation experiments were firstly performed on the Meningioma dataset, and Table 4 shows the respective experimental results based on multiple model structures.

Three commonly used image segmentation evaluation indicators are applied in the experiments for the performance evaluation, including the pixel-level F1-score which is the general evaluation of both precision and sensitivity. True Positive (TP), False Positive (FP) and False Negative (FN) of are determined by whether pixels are classified to the right or wrong predicted categories. Meanwhile, Intersection-over-Union (IoU) which is the same as JS discussed above, and Aggregated Jaccard Index (AJI) which is based on the connected domain and more

**Table 4. Ablation experimental results of multiple network structures.**

|  | Model | Patch size | AJI | Pixel-level F1 |
|---|---|---|---|---|
| Validation of global transfer connection | 3-layer UNet | 48×48 | 0.4570 | **0.8237** |
|  | 3-layer FGDC-net | 48×48 | **0.4886** | 0.8123 |
| Validation of larger sight | 4-layer UNet | 48×48 | 0.4732 | 0.7989 |
|  | FGDC-net | 96×96 | **0.5058** | **0.8169** |

**Table 5. Ablation experimental results of multiple training strategies.**

| HSV Mixup | Mixed loss | Dynamic training | AJI | Pixel-level F1 |
|-----------|-----------|------------------|-----|----------------|
|  |  |  | 0.4036 | 0.7094 |
| √ |  |  | 0.5093 | 0.8073 |
| √ | √ |  | 0.5258 | 0.8211 |
| √ | √ | √ | **0.5675** | **0.8462** |

precise than IoU are also included. The definitions of the proposed indexes are listed below.

$$\text{F1} = \frac{2TP}{2TP + FT + FN} \tag{9}$$

$$\text{AJI} = \frac{\sum_{i=1}^{N} |G_i \cap S_j^M|}{\sum_{i=1}^{N} |G_i \cup S_j^M| + \sum_{F \in U} |S_F|} \tag{10}$$

As can be seen from Table 4, when the number of layers is the same as that of UNet, FGDC-net structure is superior to UNet structure in AJI index, while pixel-level F1 is only slightly different. Based on the definition of AJI, it shows that FGDC-net structure effectively suppresses FP value, so as to minimize the occurrence of falsely predicted areas in segmentation results. In addition, although pixel-level F1 does not change significantly after the padding of training images, the AJI index is further improved by using FGDC-net. The two major improvements of the network including padding of larger training patches and introducing FGDC modules bring positive effects on the segmentation ability of the model as shown in the last row of Table 4.

In order to evaluate the effectiveness of three major optimizations in the network training, the ablation experiments based on FGDC-net model is shown in Table 5, in which BCELoss is applied as the loss function when mixing loss is not used. Similarly, when the dynamic training was not adopted, the training period epoch was set to a fixed number of 400 and the mixed probability was set as 0.5.

As shown in Table 5, after HSV Mixup operation was performed on the training samples, the performance of the model was significantly improved, indicating the HSV Mixup operation helps the network to better identify the nuclei with lighter staining, thus greatly improves the segmentation accuracy. Besides, since the participation of DiceLoss in the mixing loss, the attention mechanism of the proposed model is enhanced to detect and predict smaller nuclei, which improves the segmentation performances including AJI and F1 comparing to simply using BCELoss. The last row shows that dynamic training strategies further improve the generalization ability of the model with flexible probabilities of Full Mixup of training patches and the related changing of epochs.

## Comparison of results on open datasets

The optimized algorithm with FGDC-net structure and improved training strategies was further verified on the open-source datasets of Kumar and MoNuSeg. A batch of classic methods based on different machine learning strategies including supervised learning, weakly supervised learning, and unsupervised learning methods from literatures were included in Tables 6 and 7 respectively.

Table 6 includes several classical supervised learning models, as well as the results of weakly supervised and unsupervised learning models, and most of current weakly supervised and

**Table 6. Comparison experimental results on Kumar dataset.**

| | Method | AJI | Pixel-level F1 | IoU |
|---|---|---|---|---|
| Supervised learning | FCN [33] | 0.3556 | **0.7809** | —— |
| | Mask-Rcnn [33] | 0.5002 | 0.7470 | —— |
| | CNN3 [8] | 0.5083 | 0.7623 | —— |
| Weakly supervised learning | Qu et al.(5%) [34] | 0.4941 | 0.7540 | —— |
| | Pseudo EdgeNet [35] | —— | —— | 0.6136 |
| Unsupervised learning | SIFA [36] | 0.3924 | 0.6880 | —— |
| | CyCADA [37] | 0.4447 | 0.7220 | —— |
| | Mihir et al. [38] | 0.5354 | 0.7477 | —— |
| | DDMRL [39] | 0.4860 | 0.7109 | —— |
| Ours | | **0.5238** | 0.7655 | **0.6202** |

unsupervised learning models have lower performances than supervised deep learning models. However, due to the scarcity of labeled of segmented pathological image samples, weakly supervised and unsupervised learning models are the inevitable trend of network development. For our algorithm on Kumar dataset, AJI reaches 52.38%, which was better than the previous best CNN3 50.83%. F1 reached 76.55%, not much different from FCN, MASK-RCNN and CNN3. Generally, the proposed model belongs to the category of unsupervised learning, our method achieves best overall performances except for Pixel-level F1 index, which is slightly lower than that of FCN model. Based on the proposed algorithm, the sample segmentation results of different organs from all three datasets are shown in Fig 7, where the original image is shown in the top left of each organ block, and three local ROIs are randomly selected as orange, blue and yellow windows from the original image, and the last column in each organ block shows four zoomed areas with detailed nuclei boundaries marked as green lines.

The comparison results of MoNuSeg dataset between multiple methods are shown in Table 7. Similar to that of Table 6, the only difference is there is leak of literature on unsupervised learning methods in MoNuSeg dataset at present. For our method on MoNuSeg dataset, the pixel-level F1 index reaches 77.15%, which is more than that of classic FCN, DB-UNet, and SegNet. Another segmentation IoU index of our algorithm is also better than the previous best DB-UNet by more than two percents. The results show that FGDC-Net could effectively improve the segmentation effect of HE staining pathological images, and Fig 7 shows the partial segmentation results.

**Table 7. Comparison experimental results on MoNuSeg dataset.**

| | Method | AJI | Pixel-level F1 | IoU |
|---|---|---|---|---|
| Supervised learning | FCN [40] | 0.3510 | 0.7460 | 0.4935 |
| | UNet++ [41] | —— | 0.7453 | 0.5892 |
| | deeplabv3+ [42] | —— | 0.7185 | 0.5619 |
| | DB-UNet [43] | —— | 0.7421 | 0.6016 |
| | SegNet [44] | —— | 0.7526 | —— |
| Weakly supervised learning | BoundingBox [45] | —— | 0.7372 | 0.5839 |
| | Self-loop(20%) [46] | —— | 0.7711 | —— |
| | SSL(10%) [47] | 0.5501 | —— | —— |
| Ours | | **0.5512** | **0.7715** | **0.6297** |

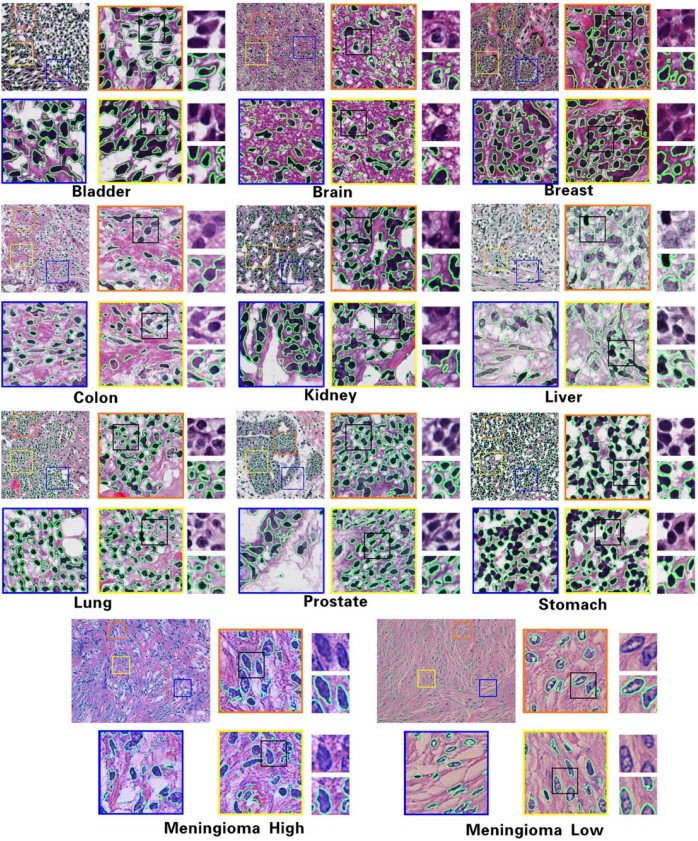

**Fig 7. Samples segmentation results of multiple organs from three datasets.**

## Conclusions

In this paper, a fully automated pipeline based on Feature Global Delivery Connection Network to locate precise nuclear boundaries in HE-stained pathological images is proposed. To achieve the deep learning-based image segmentation in a fully automatic way, the framework is further enhanced in each stage of the pipeline, including the automatic training sample generation, the new segmentation module structural design, and a flexible training strategy. First, the unsupervised training sample selection method generates extended image patches and their corresponding binary label maps, in which the certain size difference between original patches and their labels leads to the asymmetric model design of FGDC-net, and improves the segmentation performance by the attention mechanism. Meanwhile, by mixup of image patches in HSV color spaces, higher quality image patches with better nuclei pixels distributions are generated for the network training, which further improves the effectiveness of the unsupervised deep learning with higher efficiency than the supervised methods. Second, the proposed FGDC module abandons the jumping connections between codecs, which achieves feature selection by learning the relationships between feature channels at each layer and pass information selectively. The asymmetric design of FGDC-net also forms an attention mechanism by increasing the local receptive field of the central region to be segmented. Third, the probability of mixing different image patches for the training is constantly adjusted by the combination of both BCE and Dice loss functions, and the epoch of training is also affected.

These dynamic training strategies fully consider both mixed and unmixed samples, and further improve the generalization ability of the model.

Comparing to existing state-of-the-art supervised, weakly supervised and unsupervised learning methods dealing with pathological image segmentation, the proposed method shows better overall performances considering both efficiency and accuracy. Our unsupervised segmentation algorithm does not require human participation in constructing the training set, replacing the most time- and labor-intensive outlining and labeling parts of traditional deep learning, and significantly improving the efficiency of image analysis by automatically generating reliable labels for model training. The optimized method improves the accuracy of the unsupervised method to 0.7655 and 0.7715 on the publicly available datasets Kumar and MoN-uSeg respectively. It is further demonstrated that the presented algorithm significantly improves the training sample production efficiency while achieving a certain degree of improvement in segmentation accuracy.

Meanwhile, the proposed FGDC-net is designed to optimize the information transfer between codecs, which filters and integrates the shallow information and then control the importance of each layer feature as a form of weight. The modules further selectively integrate deep features and improves feature representation while preserving the information exchange between encoder and decoder. Experimental results show that the proposed segmentation method can achieve high segmentation accuracy on both clinical and public available datasets, provide more accurate feature indicators related to pathological images for cancer analysis and diagnosis, and will promote the application of automatic quantitative pathological image analysis technology in clinical aid diagnosis.

To further improve the effectiveness of deep learning on pathological image researches, it will be an inevitable trend towards weak and unsupervised development because of the scarcity of HE and other stained pathological images and labeling in deep learning. In the future, the construction of fuzzy neural network model for medical image analysis will be further studied to improve the efficiency of deep learning from both sample generation and network training. While solving the problem of low efficiency and subjectivity of manual drawing, the network optimization training is realized efficiently and controllable. Furthermore, the histological and microscopic structure of individual cells was accurately measured according to the independent cell boundary, and the correlation analysis between morphological characteristics and pathological classification will be carried out to explore the cytological mechanism corresponding to tumor lesions, and to establish a pathological imaging diagnosis and treatment model reflecting pathological heterogeneity.

## Supporting information

**S1 Fig. Original pathological images of Fig 1 in the main body.**
(ZIP)

**S2 Fig. Original pathological images of Fig 2 in the main body, and its corresponding original image.**
(ZIP)

**S3 Fig. Original pathological images of Fig 6 in the main body, and its corresponding original image.**
(ZIP)

**S4 Fig. Original pathological images of Fig 8 in the main body, and its corresponding original images of different organs.**
(ZIP)

**S5 Fig. Our own dataset including original meningioma images 1 to 12.**
(ZIP)

**S6 Fig. Our own dataset including original meningioma images 13 to 24.**
(ZIP)

**S7 Fig. Our own dataset including original meningioma images 25 to 36.**
(ZIP)

**S8 Fig. Our own dataset including original meningioma images 37 to 48.**
(ZIP)

**S9 Fig. Our own dataset including original meningioma images 49 to 60.**
(ZIP)

**S1 File. Original document of approval from local ethics committee.**
(PDF)

## Author Contributions

**Conceptualization:** Peng Shi, Jing Zhong, Lin Lin.

**Data curation:** Liyan Lin, Lin Lin.

**Formal analysis:** Jing Zhong, Liyan Lin, Lin Lin, Chongshu Wu.

**Funding acquisition:** Peng Shi, Jing Zhong.

**Investigation:** Liyan Lin, Lin Lin.

**Methodology:** Peng Shi, Jing Zhong, Huachang Li, Chongshu Wu.

**Project administration:** Peng Shi.

**Supervision:** Peng Shi.

**Writing – original draft:** Peng Shi, Jing Zhong, Huachang Li, Chongshu Wu.

**Writing – review & editing:** Peng Shi, Jing Zhong, Huachang Li, Chongshu Wu.

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
