## [Decision Letter · Decision Letter 0]

21 Jun 2022

PONE-D-22-11182Segmentation of HE Stained Histopathological Images Based on Feature Global Delivery Connection NetworkPLOS ONE

Dear Dr. Shi,

Thank you for submitting your manuscript to PLOS ONE. After careful consideration, we feel that it has merit but does not fully meet PLOS ONE’s publication criteria as it currently stands. Therefore, we invite you to submit a revised version of the manuscript that addresses the points raised during the review process. Your manuscript has been revised by three experts in the field and their reports are attached here below for your reference. The are some major concerns that need to be addressed carefully in a major round of revisions, and in particular: The title doesn't seem to reflect the content of the manuscript, as it suggests that the proposed method can segment any structure, whereas in fact only experiments for cell segmentation are reportedIt is unclear how the proposed approach improves on the existing methods. Please clarify the main points of strength of your procedure comapred with the state-of-the art.At least one image should present the result of the detected precise nuclear boundaries.The experimental design - and in particular the train/test split needs clarification.

We look forward to receiving your revised manuscript.

Kind regards,

Francesco Bianconi, Ph.D.

Academic Editor

PLOS ONE

Journal Requirements:

Reviewers' comments:

Reviewer's Responses to Questions

**Comments to the Author**

1. Is the manuscript technically sound, and do the data support the conclusions?

Reviewer #1: Yes

Reviewer #2: Yes

Reviewer #3: Partly

2. Has the statistical analysis been performed appropriately and rigorously? 

Reviewer #1: Yes

Reviewer #2: N/A

Reviewer #3: Yes

3. Have the authors made all data underlying the findings in their manuscript fully available?

Reviewer #1: Yes

Reviewer #2: Yes

Reviewer #3: Yes

4. Is the manuscript presented in an intelligible fashion and written in standard English?

Reviewer #1: No

Reviewer #2: Yes

Reviewer #3: Yes

5. Review Comments to the Author

Reviewer #1: In the paper „ Segmentation of HE Stained Histopathological Images Based on Feature Global Delivery Connection Network”, the authors proposed a method to segment structures in histological data. The main advantage of the method is the application of a few datasets, including public ones.

- the title is confusing- suggests that methods can segment any structure, whereas in the paper are presented experiments for cell-segmentation only

- lack of image that presents the result of detected precise nuclear boundaries

- the new sentence should start with a capital letter

- IOU metric is good for localization, and it is not the best for instance segmentation tasks

- How the data split of the “own dataset” was done? On a patient level?

- “ The correction-feedback process continued until the error rate of mistaken labeled nuclear pixels in each image was less than 5%, which made the labeling results dependable as the golden standards for the following training and validations.” – how the 5% value was established?

- “The correction-feedback process continued”- what type of correction was applied?

- EMC- please add an explanation for this shortcut

- it is not necessary to add formulas of a well-known metrics

- Table 6- the best results are for FCNN, in table 7 results are very close to the result for Self-loop. What advantage do we have by applying the proposed method?

Reviewer #2: An effective method named Feature Global Delivery Connection Network is developed to segment HE stained histopathology image.

I believe that papers published in PLOS ONE, should demonstrate either a new method proven by a strong theoretical background or new methods with a mature level of implementation. The method presented by the authors have a strong theoretical background. The overall idea is very interesting and the results are promising. However, I have some questions to be addressed by the authors:

1. Abstract is too long. Make it short and informative.

2. The topic is interesting and readers will be more interested to read this manuscript. The methods and ideas used in the manuscript are perfect.

3. I recommend that the author should check the manuscript properly regarding grammatical mistake, text and figure organization.

4. Check all the Figure's explanations in the body text and explain the Figure's and Table's caption in detailed and clearly.

Overall, I think this paper can be accepted after minor revision.

Reviewer #3: 1. Contribution in this paper is insignificant.

2. Results are not improved by the present method in considerable way.

3. Methods use are also not innovative.

4. Similar systems exist and without any significant gain, it is meaningless to publish another paper.

6. PLOS authors have the option to publish the peer review history of their article (what does this mean?). If published, this will include your full peer review and any attached files.

Reviewer #1: No

Reviewer #2: No

Reviewer #3: No

---

## [Author Response · Author response to Decision Letter 0]

14 Jul 2022

Reviewer #1 In the paper, Segmentation of HE Stained Histopathological Images Based on Feature Global Delivery Connection Network”, the authors proposed a method to segment structures in histological data. The main advantage of the method is the application of a few datasets, including public ones.

Question 1. the title is confusing- suggests that methods can segment any structure, whereas in the paper are presented experiments for cell-segmentation only.

Answer: As the characteristic changes of normal cells after cancerization are mostly reflected in nuclei, the method is designed mainly focusing on nuclei segmentation. To make the title more specific, it has been modified to ‘Nuclei Segmentation of HE stained Histopathological Images Based on Feature Global Delivery Connection Network’. 

Question 2. lack of image that presents the result of detected precise nuclear boundaries.

Answer: We found that the uploaded images were compressed by the submission system to make the size of combined PDF smaller, which made the resolutions of images in the final pages of generated manuscript be much lower than the original uploaded files. Therefore, the directly downloaded image files from the reviewer toolbar may be helpful to solve this problem, in which clear boundaries could be found at different magnifications. Meanwhile, we also improved the experimental segmentation results by brighter and broader lines to show clearer nuclei boundaries to replace the previous Figure 7.

Question 3. the new sentence should start with a capital letter.

Answer: We have double checked the full text and corrected these problems, which could be found in the new uploaded file.

Question 4. IOU metric is good for localization, and it is not the best for instance segmentation tasks.

Answer: Yes, IOU (Jaccard) index is a commonly used metric in object detection, which mainly reflects the overlapped area proportion between two objects. Here IOU is used to assess the localization precision of the predicted nuclear boundaries. Based on literature searching, IOU is also used as a main index in semantic segmentation, and can be found in some nuclei segmentation papers such as listed in Table 7. Therefore, we tested IOU metrics of the proposed algorithm for the comparison to existing methods especially on MoNuSeg dataset. Meanwhile, indexes which are better at dealing with under- and over-segmentation such as AJI and Pixel-level F1 were also used in evaluations of our experiments.

Question 5. How the data split of the “own dataset” was done? On a patient level?

Answer: The previous description of data split was no very clear and we have revised the paragraph accordingly. Here is the updated dataset split strategy. First, there were totally 60 cases of meningiomas blindly chosen on a patient level, in which 30 cases are from high-grade and other 30 cases are low-grade to ensure the comprehensiveness and diversity of the images. Second, one original image was randomly selected from each case, so we had 60 original images in our own dataset. Then, 10 images from each grade of cases were randomly selected as the test set, and there were totally 20 images for test. Finally, the remaining 40 images including half high-grade and half low-grade images formed the training set as shown in Table 1. The ratio between the training and test datasets is 2:1.

Question 6. The correction-feedback process continued until the error rate of mistaken labeled nuclear pixels in each image was less than 5%, which made the labeling results dependable as the golden standards for the following training and validations.” – how the 5% value was established?

Answer: Here the 5% value was an experience point, which was suggested by pathologists in our research team. Since the one of the main concerns of the proposed method is to generate emulational nuclei mask labels fully automatically to improve the effectiveness, the number of nuclei labels needed for training is quite huge. In practice, one 1536×2048 meningioma pathological image averagely contains about 800 nuclei that need to be labeled, so a total of about 16,000 nuclei needs to be labeled in 20 training images. Considering an acceptable error rate, our pathologists used 5% as the threshold of mistaken labeled nuclear pixels in checking the labeling results of MaZda software. The effectiveness of the automatic labeling was proved and described in our previous work, so we didn’t discuss this in details in this new work.

Question 7. “The correction-feedback process continued”- what type of correction was applied?

Answer: As described in Section 2.1.3, if the error rate of mistaken labeled nuclear pixels in each image was more than 5%, we checked the 5% mostly mistakenly labeled nuclei and manually correct their boundaries by pathologists, and then double check the new error rate of mistaken labeled nuclear pixels in the image. If the rate was less than 5%, the labels were considered as the golden standards of this image, and if not, the rest 5% mostly mistakenly labeled nuclei were manually corrected till the error rate of mistaken labeled nuclear pixels meted the requirement. A brief introduction of the correction-feedback process is also added into the manuscript. 

Question 8. EMC- please add an explanation for this shortcut.

Answer: Sorry we couldn’t find shortcut named EMC in the manuscript. I think the Reviewer may mean ECM and it is the shortcut of Extra Cellular Matrix, which is one of three main structures in HE stained tissues including nuclei, cytoplasm and ECM. To be consistent to our previous work, we revised ECM to Extra Cellular Space (ECS) in the new text. The explanation of ECS is in the final paragraph of Introduction. We have capitalized the first letters to make it more visible.

Question 9. it is not necessary to add formulas of a well-known metrics.

Answer: We found some of papers concerning cell segmentation included the formulas of commonly used metrics by literature searching. As suggested, we have removed the formulas of well-known metrics including BCE Loss, Dice loss and Jaccard Similarity (Equations 9 - 11) to make the descriptions more concise.

Question 10. Table 6- the best results are for FCN, in table 7 results are very close to the result for Self-loop. What advantage do we have by applying the proposed method?

Answer: Yes, FCN makes the best Pixel-level F1 index as shown in Table 6, and our method take the second place in that column. Meanwhile, it is indeed that most of results are very close to ours including the Self-loop method. The main purpose of our proposed method is to build a framework that could fully automatically segment nuclei boundaries without manual intervention. Since most of the current deep learning-based methods need labeled nuclei samples by exports for training, the automatic generation of nuclei labels greatly helps to save time and efforts of labeling in our proposed framework. With the huge increase of efficiency, the errors of mistaken labeled nuclear pixels in nuclei label generation also affect the network training and then the accuracy of segmentation. As an unsupervised learning approach, our method has better segmentation performances than most of the supervised and weakly supervised learning-based methods, in which the weakly supervised learning methods including Self-loop also need a small number of labeling works by exports. Therefore, we think the main advantage of the proposed method is making good performances in increasing the accuracy with highly improving the efficiency, which may be suitable to be applied in clinical use in the future.

Reviewer #2 An effective method named Feature Global Delivery Connection Network is developed to segment HE stained histopathology image. I believe that papers published in PLOS ONE, should demonstrate either a new method proven by a strong theoretical background or new methods with a mature level of implementation. The method presented by the authors have a strong theoretical background. The overall idea is very interesting and the results are promising. However, I have some questions to be addressed by the authors:

Question 1. Abstract is too long. Make it short and informative.

Answer: We have revised the Abstract to make it more concise and to the point. First, we deleted the less innovative step of ResBlock module, and the description of F1, IoU and AJI metrics. Second, the focus on nuclei segmentation was emphasized according to the title change. The total number of words has been reduced from 292 to 249. 

Question 2. The topic is interesting and readers will be more interested to read this manuscript. The methods and ideas used in the manuscript are perfect.

Answer: Thanks for the comments. 

Question 3. I recommend that the author should check the manuscript properly regarding grammatical mistake, text and figure organization.

Answer: We have double checked the manuscript by proof reading, and the grammatical mistakes, text and figure organization problems have been revised accordingly, especially the marked nuclei boundaries showing segmentation performances in Figure 7.

Question 4. Check all the Figure's explanations in the body text and explain the Figure's and Table's caption in detailed and clearly.

Answer: We have double checked Figure's explanations in the body text. All Figure’s and Table’s captions are explained emphatically as can be seen in the revised file.

Overall, I think this paper can be accepted after minor revision.

Reviewer #3: 

Question 1. Contribution in this paper is insignificant.

Answer: We were focusing on resolving two difficulties of current automated histopathological images analysis researches. First, from the aspect of efficiency, the lack of labeled nuclei for training of deep learning-based segmentation approaches. We used asymmetric labels and HSV color transform to improve the consistency between the automatically generated pseudo labels and real samples. Second, from the aspect of accuracy, the loss of relationships between feature channels at each layer inside the segmentation network. We designed FGDC module to use sigmoid-like gating to assign weights to intra-layer feature channels of each layer to achieve feature screening. These two contributions made the proposed method obtain good segmentation performances as most of the current supervised and weakly-supervised methods have. Meanwhile, since the deep-learning based segmentation methods are going to mature and stable in recent years, big improvements of assessment metric are difficult and experimental results of most published methods are slightly improved according to literature searching, which are also shown in Table 6 and 7.

Question 2. Results are not improved by the present method in considerable way.

Answer: We designed three groups of experiments to show the improvements of the proposed method. First, to show the effectiveness of the proposed improvements including HSV Mixup in nuclei label generation, Mixed loss in dynamic training, and the FGDC module in network structure, we compare them with our previous work by ablation experiments, and the results in Table 4 and 5 show that those improvements are necessary to increase the segmentation accuracies. Second, since the nuclei mask labels are automatically generated rather than manually marked, most of unsupervised approaches have lower segmentation accuracies than other training strategies. The comparison experiments to other methods from literatures based on open datasets further proved the proposed method have similar or better segmentation performances in a fully automatic way. 

Question 3. Methods use are also not innovative.

Answer: There are multiple strategies dealing with nuclei segmentation because of the clinical significance, and deep leaning-based methods have been in the majority in recent years. Although high improved the segmentation performances, the biggest problem of deep-learning strategies is that a large number of labeled samples are needed for training, which greatly affect the application in clinic. In order to solve this problem, the main innovation of our proposed method is to generate nuclei mask labels fully automatic to save time and efforts in training, and keep the segmentation accuracy in a high level by designing new modules for the network. Therefore, the innovative of works mainly include asymmetric labels and HSV color transform in the stage of automatic training sample patches generation, and a redesigned FGDC-net model rather than the previous U-net based segmentation framework. Experimental results on multiple organ datasets also proved the effectiveness of the proposed improvements. The corresponding discussions is in Paragraph 2 of Conclusions.

Question 4. Similar systems exist and without any significant gain, it is meaningless to publish another paper.

Answer: Comparing to other methods and our previous work, the main innovations of FGDC-net mainly include two aspects. First, in the stage of automatic training sample patches generation, asymmetric labels and HSV color transform are employed to improve the consistency between the generated pseudo labels and real samples. Second, the network model is redesigned as a FGDC-net rather than the previous U-net based segmentation framework. FGDC module is proposed to use sigmoid-like gating to assign weights to intra-layer feature channels of each layer to achieve feature screening, which increases the attentions on feature maps of nuclei segmentation. Ablation experimental results showed that the above improvements made FGDC-net further improved the segmentation performances than the previous U-net based network based on the same datasets. Then, we used FGDC-net to test on two open datasets and also achieved good performances. The above innovations in this manuscript are significantly different from other existing methods rather than a simple repeated work. The corresponding discussions is in Paragraph 2 and 3 of Conclusions.

---

## [Decision Letter · Decision Letter 1]

15 Aug 2022

Nuclei Segmentation of HE Stained Histopathological Images Based on Feature Global Delivery Connection Network

PONE-D-22-11182R1

Dear Dr. Shi,

We’re pleased to inform you that your manuscript has been judged scientifically suitable for publication and will be formally accepted for publication once it meets all outstanding technical requirements.

Within one week, you’ll receive an e-mail detailing the required amendments. When these have been addressed, you’ll receive a formal acceptance letter and your manuscript will be scheduled for publication. Please note that Reviewer #2 requested minor changes to be intorduced in the final version (see below).

Kind regards,

Francesco Bianconi, Ph.D.

Academic Editor

PLOS ONE

Reviewers' comments:

Reviewer's Responses to Questions

**Comments to the Author**

1. If the authors have adequately addressed your comments raised in a previous round of review and you feel that this manuscript is now acceptable for publication, you may indicate that here to bypass the “Comments to the Author” section, enter your conflict of interest statement in the “Confidential to Editor” section, and submit your "Accept" recommendation.

Reviewer #2: All comments have been addressed

2. Is the manuscript technically sound, and do the data support the conclusions?

Reviewer #2: Yes

3. Has the statistical analysis been performed appropriately and rigorously? 

Reviewer #2: N/A

4. Have the authors made all data underlying the findings in their manuscript fully available?

Reviewer #2: Yes

5. Is the manuscript presented in an intelligible fashion and written in standard English?

Reviewer #2: Yes

6. Review Comments to the Author

Reviewer #2: This manuscript proposed a framework for nuclei segmentation. Comparison to the other state-of-the-art methods shows the superiority of the proposed method.

I cannot see what else could be added at this point - maybe with just one exception: What is your strategy for making this available for the target group (pathologists)? Please discuss this subject in your paper for better understanding. Over all study looks interesting.

I have seen the changes made to the paper, and I agree with its publication after correcting the errors:

(1) Neclei (change to Nuclei) Segmentation of HE Stained Histopathological Images Based on Feature Global Delivery Connection Network.

(2) Please check line number 214. It is confusing.

7. PLOS authors have the option to publish the peer review history of their article (what does this mean?). If published, this will include your full peer review and any attached files.

Reviewer #2: **Yes: **Subrata Bhattacharjee

---

## [Editor Report · Acceptance letter]

19 Aug 2022

PONE-D-22-11182R1 

Neclei Segmentation of HE Stained Histopathological Images Based on Feature Global Delivery Connection Network 

Dear Dr. Shi:

I'm pleased to inform you that your manuscript has been deemed suitable for publication in PLOS ONE. Congratulations! Your manuscript is now with our production department. 

Kind regards, 

on behalf of

Prof. Francesco Bianconi 

Academic Editor

PLOS ONE